# Exome Sequencing Reveals Biallelic Mutations in *MBTPS1* Gene in a Girl with a Very Rare Skeletal Dysplasia

**DOI:** 10.3390/diagnostics14030313

**Published:** 2024-01-31

**Authors:** Víctor Raggio, Soledad Rodríguez, Sandra Feder, Rosario Gueçaimburú, Lucía Spangenberg

**Affiliations:** 1Departamento de Genética, Facultad de Medicina, Universidad de la República, Montevideo 11800, Uruguay; vraggio@fmed.edu.uy (V.R.); soledadrmb@gmail.com (S.R.); 2Laboratorio de Genética Clínica Genodiagnosis, Montevideo 11600, Uruguay; sfeder@genodiagnosis.com; 3Centro de Referencia Nacional de Defectos Congénitos y Enfermedades Raras (CRENADECER), Av. Agraciada 2989, Montevideo 11800, Uruguay; dra.rosario@gmail.com; 4Hospital Británico, Av. Italia 2420, Montevideo 11600, Uruguay; 5Departamento Básico de Medicina, Hospital de Clínicas, Facultad de Medicina, Universidad de la República, Montevideo 11600, Uruguay; 6Bioinformatics Unit, Institut Pasteur de Montevideo, Montevideo 11400, Uruguay

**Keywords:** molecular genetics, massive sequencing, medical genomics, osseous dysplasias, rare diseases, diagnostic odyssey

## Abstract

The Kondo-Fu type of spondyloepiphyseal dysplasia (SEDKF) is a rare skeletal dysplasia caused by homozygous or compound heterozygous mutations in the *MBTPS1* gene. The *MBTPS1* gene encodes a protein that is involved in the regulation of cholesterol and fatty acid metabolism. Mutations in *MBTPS1* can lead to reduced levels of these lipids, which can have a number of effects on development, including skeletal anomalies, growth retardation, and elevated levels of blood lysosomal enzymes. This work reports the case of a 5-year-old girl with SEDKF. The patient had a severely short stature and a number of skeletal anomalies, including kyphosis, pectus carinatum, and reduced bone mineral density. She also had early onset cataracts and inguinal hernias. Genetic testing revealed two novel compound heterozygous variants in the *MBTPS1* gene. These variants are predicted to disrupt the function of the MBTPS1 protein, which is consistent with the patient’s clinical presentation. This case report adds to the growing body of evidence that mutations in the *MBTPS1* gene are causal of SEDKF. We summarized the features of previous reported cases (with age ranges from 4 to 24 years) and identified that 80% had low stature, 70% low weight, 80% had bilateral cataracts and 70% showed Spondyloepiphyseal dysplasia on X-rays. The findings of this study suggest that SEDKF is a clinically heterogeneous disorder that can present with a variety of features. Further studies are needed to better understand the underlying mechanisms of SEDKF and to develop more effective treatments.

## 1. Introduction

The Kondo-Fu type of spondyloepiphyseal dysplasia (SEDKF) is a recessively inherited skeletal dysplasia consisting in a phenotype that includes severe growth retardation and skeletal anomalies, including spondyloepiphyseal dysplasia with associated kyphosis, pectus carinatum and reduced bone mineral density, minor dysmorphic features, and, notably, early onset cataracts and inguinal hernias. Also, elevated levels of blood lysosomal enzymes might be present [1].

In 2018 SEDKF was found to be caused by biallelic mutations in the membrane-bound transcription factor protease, site-1 encoded by *MBTPS1* gene (16q23.3-q24.1). This protein appears to have a central role in cellular lipid metabolism. Since this first published association of *MBTPS1* gene with a human disease [1] a few additional disease reports have been published to date [2]. The first case evidenced the role of *MBTPS1* gene with skeletal development in a pediatric patient with a rare skeletal dysplasia and elevated blood lysosomal enzymes. They identified compound heterozygous variants in *MBTPS1*; one missense variant (NM_003791.3: c.1094A>G; D365G) and one 1bp duplication creating a nonsense mutation (NM_003791.3: c.285dupT; p.D96X) causing about 1% of the expected functional *MBTPS1* transcripts causing a newly identified phenotype, the SEDKF [1]. The second case reported a patient with the previously reported homozygous missense variant with a “Silver-Russell syndrome” [2], actually a different entity which shares many features with SEDKF.

The lipid composition of animal cells is controlled by SREBPs (sterol regulatory element binding proteins), membrane-bound transcription factors that are key regulators of cholesterol and fatty acid biosynthesis and uptake [3]. SREBPs are synthesized in the endoplasmic reticulum (ER). Low levels of intracellular cholesterol activates SREBPs translocation to the Golgi, where they are consecutively cleaved by S1P (site-1 protease) and S2P (site-2 protease) for activation. Active SREBPs translocate to the nucleus where it binds to cis-acting elements required for cholesterol biosynthesis, hence restoring cholesterol levels [1]. S1P interacting protein is encoded by membrane-bound transcription factor peptidase, site 1 called *MBTPS1* gene. S1P is expressed in the Golgi where it interacts with S2P, encoded by the membrane-bound transcription factor peptidase, site 2, the *MBTPS2* gene. Also, S1P functions independently from S2P by activating GlcNAc-1-phosphotransferase (GPT) as shown by an in vitro study [4]. GPT is required for the mannose-6-phosphate (M6P) modification of lysosomal enzymes in the Golgi organelle, so their transport to the lysosomes can be achieved via M6P receptors [4]. Problems in such modifications impact lysosomal transport causing accumulation of those proteins in lysosomes, hence enabling lysosomal diseases [4,5].

In vivo studies in mice show reduced biosynthesis of cholesterol and fatty acids in livers with disrupted *MBTPS1* [6]. Further studies have shown an involvement of S1P in skeletal development in zebrafish [7] and in mice [8,9]. 

Whole exome sequencing (WES) has emerged as a cost-effective approach for investigating rare diseases, given that 80% of these conditions are associated with coding genes [10]. While whole genome sequencing provides more extensive information, its current cost remains prohibitive for many countries, and the associated analysis can be challenging and computationally intensive.

Here, we report the case of a 4 years old girl with a severely short stature and skeletal anomalies that turned out to be a SEDKF as revealed by WES. We report two novel compound heterozygous variants and confirm as far as we know the 7th case (2018–2023) [1,2,11,12,13] world-wide of SEDKF confirmed at molecular level.

## 2. Materials and Methods

### 2.1. Sample Extraction and Sequencing

At first, exome sequencing was performed in the index case and then completed with trio analysis including her parents. DNA was extracted from whole blood. The index case exome was done with a library preparation that included following kits: xGen Exome Research panel, xGen CNV backbone panel and xGen Human mtDNA Research Panel (IDT, USA/Canada, according to manufacturer instructions). Whole exome sequencing was done with 100× in a NextSeq Illumina (Illumina, San Diego, CA, USA) sequencing machine in DASA (Brasil). 

Parents DNA was prepared using an Agilent V6 Library kit (according to manufacturer’s instructions) and were sequenced at 100× in a HiSeq2500 Illumina sequencing machine in Macrogen (Seoul, Republic of Korea).

### 2.2. Bioinformatics Pipeline, Variant Detection and Interpretation

The index case was analyzed in the Illumina platform Dragen and with the Edmegene platform in a commercial setting. The following rationale was used for variant filtration: coding (i) heterozygous variants, (ii) homozygous variants, and (iii) compound heterozygosity variants, all with low frequency. As a result, the company reported two variants in the *MBTPS1* gene.

With that information we analyzed the WES of the parents.

Both parents were quality processed with FastQC [14,15], and reads were mapped with BWA [16] using human genome version GRCh37. Variant calling was done with GATK [17] according to best practices. Data annotation was done with ANNOVAR [18]. 

To assess potentially causative variants that could affect their offspring recessively we filter variants according to following criteria: heterozygous mutations in coding/splicing regions with a population frequency less than 0.5%, that are present in both parents on the same gene.

## 3. Results

### 3.1. Case Report

The index case is a five years old girl. She has no family history worth mentioning and her parents were not consanguineous. She has a healthy sister. Her mother had pregnancy-related arterial hypertension and possibly preeclampsia. She was born at 36 weeks of gestation by cesarean section due to intrauterine growth restriction of unknown cause. No other abnormalities were observed at that time. Neonatal screening showed no increased risk for the disorders tested (aminoacidopathies, congenital hypothyroidism, cystic fibrosis, medium chain acyl-CoA dehydrogenase, organic acidemias and fatty acid β-oxidation disorders). 

Since birth, her growth was always under 3rd percentile. Her neurological development was normal. She presented with a triangular face with frontal bossing but no other facial dysmorphic features. Her neck was slightly short and a pectus carinatum was present albeit discrete. Her limbs were shortened at the proximal level and 5th finger clinodactyly was present in both hands. Her second toe was over the third and a reducible pied bot was present.

At first a diagnosis of Silver-Russell syndrome was proposed, studies oriented to detect uniparental disomy of chromosome 7 or methylation errors (loss of methylation of the imprinting center 1 region (H19/IGF2:IG-DMR)) of region 11q15.15, as well as karyotype, which were all normal.

At age two she underwent surgical correction of bilateral inguinal hernia and subsequently bilateral cataracts were diagnosed. To this point the diagnosis of SSR was challenged and an exome sequencing study was performed as discussed below.

Upon clinical reexamination at four years old she presented no skin lesions or alopecia areata, but her hair was sparse and thin. Facial and corporal characteristics are the same as previously described. Neuropsychological development was normal.

Her echocardiogram, cranial magnetic resonance, fundus examination and abdominal ultrasonography were all normal. Her thyroid hormones were all normal. Cystic fibrosis was ruled out. Her plasma lysosomal enzymes could not be analyzed.

As she grew older, she complained of muscular fatigability and back pain. At age five, spinal and pelvic radiographic analysis showed discrete enlargement and flattening of both femoral heads, with shortening of femoral necks, and grade I listhesis in L5-S1, in T7-T8 and L5-S1 there is a decrease in the height of the intervertebral spaces and irregularity of the vertebral plates, with images compatible with Schmôrl’s nodules associated with spinal edema in that sector (Figure 1). No acetabular morphological anomalies, osteonecrosis or signs of hip dysplasia were noticed. The Osseous signal intensity was normal. No discal hernias, canal stenosis, spinal or nerve compression were present. No significant anomalies were noted in the cervical column. 

### 3.2. Bioinformatic Analysis Revealed Pathogenic Compound Heterozygous Mutation in MBTPS1 Gene

We did the WES (100×), 100 bp paired-end sequencing on both progenitors. We obtained 120,237 variants in the mother and 118,163 in the father, from which 543 and 490 were heterozygous exonic/splicing with frequency less than 0.5%, respectively in mother and father. Those variants were shared on 28 genes in both parents. Among those genes we found the variants in the *MBTPS1* gene, coherent with the patient’s phenotype. No other variants potentially related to the phenotype were found.

The patrilineal variant (Figure 2A, bottom) corresponds to a novel heterozygous frameshift deletion p.Met785fs (c.2355delG), that falls between 46% and 75% of the protein depending on the transcript. It is predicted to cause NMD (nonsense-mediated decay) and has a “neutral” pathogenic classification as predicted by SiftINDEL. In this context, ‘neutral’ is one of the two classes that SiftIndel is trained to classify (neutral and damaging) [19].

The matrilineal variant (Figure 2B, top) corresponds to an novel heterozygous missense variant p.Asp365Gly (c.A1094G), not reported in databases, with pathogenic in silico predictions (SIFT, LRT, MutationTaster) and a CADD Phred score of 16.

Sanger sequencing was done in the patient and both variants were present meaning that we confirmed the compound heterozygous state of both variants. 

## 4. Discussion

To our knowledge this is the 7th case published so far with a diagnosis of SEDKF and the 10th with *MBTPS1* related phenotypes, hence this case report provides data for further delineation of this disease and its mutational landscape. Table 1 shows previously reported cases with their clinical features and reported mutations.

As in other cases [2] the child was at first misdiagnosed as Silver-Russell syndrome. Actually until the onset of cataracts at 2 years old the clinical picture was one of a typical SRS and the use of the Netchine-Harbison score [23] “confirmed” this diagnosis. The onset of cataracts challenged this diagnosis and prompted the realization of exome sequencing, which guided the diagnosis by the finding of two variants in *MBTPS1*. Upon clinical reanalysis the diagnosis was highly concordant with the patient’s phenotype. 

As in other cases the main findings were severe, prenatal onset and no catch-up short stature, prominent forehead with triangular face (which in conjunction with the previous feature prompted an initial diagnosis of SRS), pectus carinatum, bilateral inguinal hernias and early onset bilateral cataracts. Radiographic images showed discrete epiphyseal anomalies (shortening of femoral necks) and spinal defects (mostly in the lumbo-sacral region). Lysosomal enzymes could not be tested. 

As few patients received rhGH therapy an evaluation of its utility is not possible. The patient with SEDKF, reported by Kondo et al. in 2018, received growth hormone replacement therapy for only one year and was discontinued because of a limited response. A Chinese patient received rhGH therapy at a dosage of 0.15–0.2 IU/kg/d ih for 3 years (despite the child’s IGF-1 level was normal), in addition to basic therapies (calcium, vitamin A, and vitamin D) [22]. A moderate gain in stature (from −4.86SD to −3.96SD) was observed in this case. As our patient did not receive rGH treatment and is not expected to receive it, since there is no clear evidence of utility and the known fact that skeletal dysplasias tend to respond inefficiently to rGH, we cannot provide data in this regard.

In our view the genetic variants found, and the fact of these being in trans state, confirm the diagnosis of SEDKF. One of the variants found by us (the missense one) has already been reported (https://www.ncbi.nlm.nih.gov/clinvar/variation/625458/ accessed on 10 September 2023) in the original publication describing the gene-phenotype associations [1] and in other patient in homozygous state [2]. The frameshift variant is novel. Evidence of pathogenicity is enough to classify both of them as “probably pathogenic” based on ACMG, 2015 rules [24]: i. c.2355delG:p.Met785fs fulfills the following rules: PVS1 Null variant (frameshift) in a gene where loss of function (LOF) is a known mechanism of disease + PM2 Absent from controls (or at extremely low frequency if recessive) + PM4 Protein length changes due to in-frame deletions/insertions in a non-repeat region + PP4 Patient’s phenotype or family history is highly specific for a disease with a single genetic etiology; ii. c.A1094G:p.Asp365Gly fulfills: PM2 Absent from controls (or at extremely low frequency if recessive) + PM3 For recessive disorders, detected in trans with a pathogenic variant + PP3 Multiple lines of computational evidence support a deleterious effect on the gene or gene product + PP4 Patient’s phenotype or family history is highly specific for a disease with a single genetic etiology.

Almost all cases reported included variants in homozygous or compound heterozygous state and most likely implying a loss of function mechanisms for the *MBTPS1* gene. Only one case had a heterozygous de novo variant (Table 1), compatible with a gain of function mechanism. Resulting phenotypes are somehow diverse.

Additionally, we included two phenotypes in Table 1 that are related to biallelic *MBTPS1* mutations, hence overlapping features are expected. SEDKF and CAOP share the bilateral cataracts feature and also uncommon hair characteristics: CAOP has alopecia or sparse hair as associated clinical features and the SEDKF case presented here also has the sparsity of hair.

This case highlights the utility of exome sequencing combined with clinical and evolutive reanalysis in the diagnosis of rare osseous dysplasias and ultra rare and “new” diseases. 

## Figures and Tables

**Figure 1 diagnostics-14-00313-f001:**
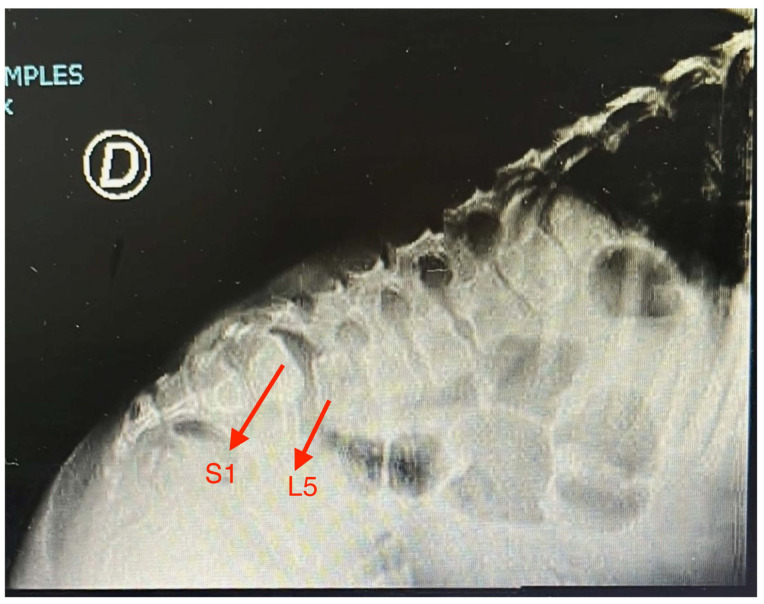
Radiographic image of lumbar sector of spine. Grade I listhesis in L5-S1 and decrease in the height of the intervertebral spaces and irregularity of the vertebral plates in L5-S1 can be seen.

**Figure 2 diagnostics-14-00313-f002:**
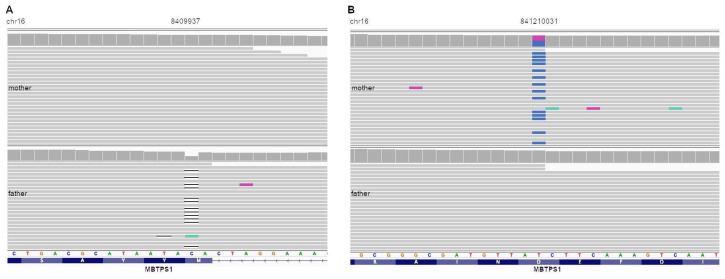
IGV screenshot of variants in both parents. Horizontal gray bars represent reads and vertical gray bars represent depth of coverage. Color portions or lines represent mutations or indels. In blue are the Guanines, in violet the Thymines, in green Adenines. Gray color in horizontal bars represent no mutations. (**A**) Frameshift variant in position 8409937 present only in the father (bottom). Depth of coverage is lower at that position, evidencing heterozygous state. (**B**) Missense variant at position 841210031 present only in the mother (top), the change is represented at that position with the blue short horizontal bars (in gray the Adenines and in blue the Guanines).

**Table 1 diagnostics-14-00313-t001:** Patients described so far with phenotypes associated with biallelic MBTPS1 pathogenic or likely-pathogenic variants.

	Yuji Kondo et al., 2018 [1]	Schweitzer et al., 2019 [20]	Meyer et al., 2020 [2]	Carvalho et al., 2020 [13]	Chen et al., 2022 (Patient 1) [21]	Chen et al., 2022 (Patient 2) [21]	Alotaibi et al., 2022 [12]	Yuan et al., 2022 [11]	Chen et al., 2023 [22]	Our Case	YES %
	PMID: 30046013	PMID: 31070020	PMID: 32857899	PMID: 32420688	PMID: 35362222	PMID: 35362222	PMID: 36330313	PMID: 36714646	PMID: 36816387	NA	
Sex	F	F	M	F	M	F	F	M	M	F	60 (F)
Age (years)	8	24	7	5	14	5	10	12	6	4	
Small for gestational age	Yes	No	Yes	Yes	NA	NA	Yes	No	No	Yes	50
Short stature	Yes	No	Yes	Yes	Yes	No	Yes	Yes	Yes	Yes	80
Low weight	Yes	No	Yes	Yes	NA	NA	Yes	Yes	Yes	Yes	70
Spondyloepiphyseal dysplasia on X-ray	Yes	NA	No	Yes	Yes	No	Yes	Yes	Yes	Yes	70
Prominent forehead	Yes	No	No	Yes	No	No	Yes	No	Yes	Yes	50
Micrognathia	No	No	No	Yes	No	No	Yes	Yes	Yes	Yes	50
Large and/or dysplastic ears	Yes	No	Yes	Yes	No	No	Yes	Yes	Yes	Yes	70
Bilateral cataracts, early onset	Yes (2 yo)	No	Yes (adolescence)	Yes (10 mo)	Yes (?)	Yes (?)	No	Yes (2 yo)	Yes (2 yo)	Yes (2 yo)	80
Pectus carinatum	Yes	No	No	Yes	NA	NA	Yes (mild)	Yes	No	Yes	50
Pectus excavatum	No	No	No	No	NA	NA	No	No	Yes	No	10
Inguinal hernias	Yes	No	Yes	No	No	No	No	Yes	No	Yes	40
Low bone mineral density	No	No	No	Yes (radiography)	NA	NA	Yes	Yes (radiography)	Yes (radiography)	Not assessed	40
Anterolisthesis of L5 on S1	Yes	No	No	No	No	No	No	No	No	Yes	20
Kypho-scoliosis	Yes	No	No	Yes	NA	NA	Yes	Yes (mild)	Yes	No	50
Chronic back pain	Yes	No	No	No	No	No	No	No	No	Yes	20
Bilateral shortening of femoral necks	Yes	No	No	Yes	NA	NA	No	No	No	Yes	30
Irregular and dysplastic appearance of femoral epiphyses	Yes	No	No	Yes	NA	NA	Yes	Yes	No	Yes	50
Irregular and dysplastic appearance of proximal tibial epiphyses	Yes	No	No	Yes	NA	NA	No	No	No	No	20
Gracile fibulae	Yes	No	No	No	NA	NA	Yes	No	No	No	20
Valgus bowing of tibiae	Yes	No	No	No	NA	NA	Yes	No	No	No	20
Defective endochondral ossification	Yes	No	No	No	NA	NA	No	No	No	No	10
Delayed ossification of epiphyses	No	No	No	No	NA	NA	No	No	No	No	0
Delayed ossification of carpal bones	No	No	No	No	NA	NA	No	No	Yes	Not assessed	10
Brachydactyly	No	No	No	No	NA	NA	Yes	No	No	Yes	20
Shortening of tubular bones	No	No	No	Yes	NA	NA	Yes	No	No	Yes	30
Delayed gross motor milestones	Yes	No	No	Yes	No	Yes	Yes	No	No	No	40
Elevated lysosomal enzymes	Yes	No	Yes	Yes	No	No	Yes	Not assessed	Not assessed	Not assessed	40
HyperCKemia	No	Yes	No	No	No	No	No	No	No	No	10
Focal myoedema	No	Yes	No	No	No	No	No	No	No	No	10
Cutaneous lesions	No	No	No	No	Yes (psoriasis, ichthyosis)	Yes (psoriasis, ichthyosis)	No	No	No	No	20
Mucosal lesions	No	No	No	No	Yes	No	No	No	No	No	10
Sparse hair	No	No	No	No	Yes	Yes	No	No	No	Yes	30
Consanguinity	No	Not assessed	Yes (distantly related)	Yes (second degree cousins)	No	No	Yes (cousins)	No	No	No	30
Associated genetic variant(s) (NM_003791.3)	c.285dupT, p.(Asp96X)/c.1094A-G, p.(Asp365Gly)	c.3007C>T, p.(Pro1003Ser)	c.1094A>G, p.(Asp365Gly)	c.2948G>A, p.(Trp983ter)	c.1064T>G, p.(Val355Gly)/c.3157T>C, p.(Ter1053Arg)	c.2072-2A>T, p.(Ter1053Cys)	c.2634C>A, p.(Ser878Arg)	c.2656C>T, p.(Q886 *)/c.774C>T, p.(A258=)	c.1589A>G, p.(Asp530Gly)/c.163T > C, p.(Glu55Lys)	c.2355delG, p.(Met785fs)/c.1094A>G, p.(Asp365Gly)	
Zigocity/inheritance	compound heterozygous	heterozygous, de novo	Homozygous	Homozygous	compound heterozygous	compound heterozygous	Homozygous	compound heterozygous	compound heterozygous	compound heterozygous	
Splicing, frameshift or stop codon	2	0	0	2	1	2	0	2	0	1	
Missense	0	1	2	0	1	0	2	0	2	1	
Functional studies on variants	Yes	Yes	No	No	Yes	Yes	No	Yes	No	No	
Initially Misdiagnosed as Silver–Russell syndrome	Yes	No	Yes	No	No	No	No	No	No	Yes	30
Final Diagnosis	SEDKF	complex phenotype	SEDKF	SEDKF	CAOP	CAOP	SEDKF	SEDKF	SEDKF	SEDKF	

CAOP: cataract, alopecia, oral mucosal disorder, and psoriasis-like. SEDKF: Spondyloepiphyseal dysplasia, Kondo-Fu type. NA: not assessed. Percentages of core features: 80% short stature, 70% low weight, 70% show Spondyloepiphyseal dysplasia on X-ray images, 70% had large and/or dysplastic ears and 80% had bilateral cataracts, early onset. In some cases the age of onset is unknown, hence the “?” sign. The * sign is used as part of variant nomenclature.

## Data Availability

Raw data is available upon request. These data are retained at the Institut Pasteur de Montevideo and cannot be made openly accessible because of ethical and privacy concerns.

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
