# Peer review of "Exome Sequencing Reveals Biallelic Mutations in MBTPS1 Gene in a Girl with a Very Rare Skeletal Dysplasia"

_diagnostics, 2024, doi:10.3390/diagnostics14030313_

Round 1
Reviewer 1 Report
Comments and Suggestions for Authors
The authors presented an intriguing case involving biallelic changes in the MBTPS1 gene, confirming a diagnosis of Kondo-Fu type of spondyloepiphyseal dysplasia. Overall, the case was well-documented and the genetic data provided solid support for the diagnosis. However, there are some questions and revisions that the authors should address before publication.
General:
• It would be helpful to include line numbers for easier reference during review and communication.
• Please ensure that proper nomenclature is used to describe the mutations.
Introduction:
• In the abstract, summarize the core features of SEDKF based on Table 1. This will make the abstract more informative and engaging to readers.
• In the last paragraph of the introduction, it is mentioned as the 6th case, but according to Table 1, it appears to be the 9th case. This is confusing and should be clarified.
Methods:
• In section 2.2, the statement "affect their offspring recessively" suggests a focus only on recessive disorders. The variant analysis flow should be more comprehensive and encompass other possibilities.
Results:
• In section 3.1, the age of the index case is stated as five years old, while in the abstract it is mentioned as a four-year-old. Please ensure consistency.
• Regarding the statement "by cesarean section due to intrauterine growth restriction of unknown cause," provide information on when the IUGR was detected prenatally, any fetal growth details, and if any other structural abnormalities were observed.
• In section 3.2, use the full term "100bp pair-end sequencing" instead of "100 PE."
• When providing the MBTPS1 variant (e.g., "MBTPS1:NM_003791:exon18:c.2355delG:p.Met785fs"), please use the appropriate nomenclature rather than a direct copy from the annotation.
• The term "neutral" pathogenic classification requires clarification. Please explain its meaning.
Discussion:
• In terms of the number of reported cases, please be consistent in stating whether it is the 6th or 7th case.
• In the sentence "A moderate gain in stature (from −4.86SD to −3.96SD)," change the font to match the rest of the text.
• Instead of "Massive sequencing," revise it as "exome sequencing" to align with the title.
Figure 1.
• Use arrows or other markers to indicate abnormalities seen in Figure 1.
Figure 2
• Is there any IGV information from patients or Sanger validation results available for Figure 2?
Table 1:
• In row 2, please include the PMID instead of the link, or use reference IDs inserted in each cell of row 1.
• Calculate the frequency of each category presented in this small cohort as a percentage and summarize which phenotypes are most common or core for MBTPS1-related Kondo-Fu type of spondyloepiphyseal dysplasia.
• Discuss the cases with the de novo variant c.3007C>T, p.(Pro1003 Ser). Do the authors believe this de novo change be a genetic diagnosis for the case in the publication? If not, consider removing it from the table and mentioning it only in the discussion section.
• In the last row, where both CAOP and SEDKF are mentioned as final diagnoses, discuss the underlying genotype-phenotype correlation.
Comments on the Quality of English LanguageThe quality of english language is generally well.
Author Response
General:
• It would be helpful to include line numbers for easier reference during review and communication.
Answer: We agree, we added line numbers.
• Please ensure that proper nomenclature is used to describe the mutations.
Answer: We checked nomenclature and corrected when appropriate.
Introduction:
• In the abstract, summarize the core features of SEDKF based on Table 1. This will make the abstract more informative and engaging to readers.
Answer: We added following line to the abstract.
We summarized the features of previous reported cases (with age ranges from 4 to 24 years) and identified that 80% had low stature, 70% low weight, 80% had bilateral cataracts and 70% showed Spondyloepiphyseal dysplasia on X-rays.
• In the last paragraph of the introduction, it is mentioned as the 6th case, but according to Table 1, it appears to be the 9th case. This is confusing and should be clarified.
Answer: We corrected the numbers in the introduction and discussion. It is the 7th SEDKF case and the 10th case of a phenotype related to that gene.
Methods:
• In section 2.2, the statement "affect their offspring recessively" suggests a focus only on recessive disorders. The variant analysis flow should be more comprehensive and encompass other possibilities.
Answer: We have rephrased the section for clarification.
Index case was analyzed in the Illumina platform Dragen and with the Edmegene platform in a commercial setting. Following rationale was used for the variant filtration: coding i) heterozygous variants, ii) homozygous variants, and iii) compound heterozygosity variants, all with low frequency. As a result, the company reported two variants in the MBTPS1 gene.
With that information we analyzed the WES of the parents.
Both parents were quality processed with FastQC13,14, and reads were mapped with BWA15 using human genome version GRCh37. Variant calling was done with GATK 16 according to best practices. Data annotation was done with ANNOVAR17.
To assess potentially causative variants that could affect their offspring recessively we filter variants according to following criteria: heterozygous mutations in coding/splicing regions with a population frequency less than 0.5%, that are present in both parents on the same gene.
Results:
• In section 3.1, the age of the index case is stated as five years old, while in the abstract it is mentioned as a four-year-old. Please ensure consistency.
Answer: We corrected the age in the abstract.
• Regarding the statement "by cesarean section due to intrauterine growth restriction of unknown cause," provide information on when the IUGR was detected prenatally, any fetal growth details, and if any other structural abnormalities were observed.
Answer: We don’t have this information and the sonograms from gestation are not available. her parents don’t remember exactly when the restriction started, but it was presumably around 30 weeks of gestation. No other abnormalities were observed at that time.
We added in the text: No other abnormalities were observed at that time.
• In section 3.2, use the full term "100bp pair-end sequencing" instead of "100 PE.”
Answer: we changed this as suggested.
• When providing the MBTPS1 variant (e.g., "MBTPS1:NM_003791:exon18:c.2355delG:p.Asp365Gly"), please use the appropriate nomenclature rather than a direct copy from the annotation.
Answer: We corrected the nomenclature and change it to: p.Asp365Gly (c.2355delG). The same for the other variant p.Asp365Gly (c.A1094G).
• The term "neutral" pathogenic classification requires clarification. Please explain its meaning.
Answer: We added following line in the text, with the corresponding citation to the article.
In this context, ‘neutral’ is one of the two classes that SiftIndel is trained to classify (neutral and damaging)18
Discussion:
• In terms of the number of reported cases, please be consistent in stating whether it is the 6th or 7th case.
Answer: We corrected this in the text. It is the 7th case and we corrected this in the introduction and the discussion.
• In the sentence "A moderate gain in stature (from −4.86SD to −3.96SD)," change the font to match the rest of the text.
Answer: Done.
• Instead of "Massive sequencing," revise it as "exome sequencing" to align with the title.
Answer: Done.
Figure 1.
• Use arrows or other markers to indicate abnormalities seen in Figure 1.
Answer: Done
Figure 2
• Is there any IGV information from patients or Sanger validation results available for Figure 2?
Answer: We were not able to obtain the raw data of the patient. This was done in a private company and we are not having success in the obtention of the fastq/BAM data. However, we do have the report of that company and the mutations found there are the ones found in the WES of the parents (one in each parent) so we trust the results of the WES of the patient. We didn’t want to take a new sample from the child to perform Sanger sequencing, since she is only 5 and has had a lot medical interventions.
Table 1:
• In row 2, please include the PMID instead of the link, or use reference IDs inserted in each cell of row 1.
Answer: Done.
• Calculate the frequency of each category presented in this small cohort as a percentage and summarize which phenotypes are most common or core for MBTPS1-related Kondo-Fu type of spondyloepiphyseal dysplasia.
Answer: We added a column to table 1 with the percentages of the clinical features of this small cohort. We also added the following to the figures caption (and also it is mentioned in the abstract):
Percentages of core features: 80% short stature, 70% low weight, 70% show Spondyloepiphyseal dysplasia on X-ray images, 70% had large and/or dysplastic ears and 80% had bilateral cataracts, early-onset.
• Discuss the cases with the de novo variant c.3007C>T, p.(Pro1003 Ser). Do the authors believe this de novo change be a genetic diagnosis for the case in the publication? If not, consider removing it from the table and mentioning it only in the discussion section.
Answer: We left the de novo case in the table since the gene is the same and it might have value for future cases. However, we added a discussion on the mechanism:
Almost all cases reported included variants in homozygous or compound heterozygous state and most likely implying a loss of function mechanisms for the MBTPS1 gene. Only one case had an heterozygous de novo variant (Table 1), compatible with a gain of function mechanism. Resulting phenotypes are somehow diverse.
• In the last row, where both CAOP and SEDKF are mentioned as final diagnoses, discuss the underlying genotype-phenotype correlation.
Answer: We added following paragraph to the discussion.
Additionally, we included two phenotypes in Table 1 that are related to biallelic MBTPS1 mutations, hence overlapping features are expected. SEDKF and CAOP share the bilateral cataracts feature and also uncommon hair characteristics: CAOP has alopecia or sparse hair as associated clinical features and the SEDKF case presented here has also the sparsity of hair.
Reviewer 2 Report
Comments and Suggestions for Authors
1. Figure 1, x ray should improve the quality and highlight the indicated information.
2. Introduction and discussion about why choose Exome sequencing, but not whole genome sequencing should be added.
Author Response
- Figure 1, x ray should improve the quality and highlight the indicated information.
Answer: we added arrows to indicate the information in the figure 1.
2. Introduction and discussion about why choose Exome sequencing, but not whole genome sequencing should be added.
Answer: We added following paragraph to the introduction.
Whole exome sequencing (WES) has emerged as a cost-effective approach for investigating rare diseases, given that 80% of these conditions are associated with coding genes10. While whole genome sequencing provides more extensive information, its current cost remains prohibitive for many countries, and the associated analysis can be challenging and computationally intensive.